# Understanding the Transfer of High-Level Reinforcement Learning Skills Across Diverse Environments

## Abstract

A large number of reinforcement learning (RL) environments are available to the research community. However, due to differences across these environments, it is difficult to transfer skills learnt by a RL agent from one environment to another. For this transfer learning problem, a multitask RL perspective is considered in this paper, the goal being to transfer the skills from one environment to another using a single policy. To achieve such goal, we design an environment agnostic policy that enables the sharing of skills. Our experimental results demonstrate that: (a) by training on both desired environments using standard RL algorithms, the skills can be transferred from one environment to another; (b) by changing the amount of data that the RL algorithm uses to optimize the policy and value functions, we show empirically that the transfer of knowledge between different environments is possible, and results in learning tasks with up to 72% fewer gradient update steps. This study takes an important step towards enabling more effective transfer of skills by learning in multitask RL scenarios across diverse environments by designing skill-sharing, sample-efficient RL training protocols.

## 1 Introduction

There have been many successes in recent years where Reinforcement Learning (RL) has shown performance on par or above humans, in domains such as video games (Mnih et al. (2015), Silver et al. (2016)), continuous control (Levine et al., 2016), chip design (Mirhoseini et al., 2021), controlling nuclear fusion (Degrave et al., 2022), and even interacting with humans (Liu et al., 2023). However, these learned policies are only effective for the task and domain they were trained in and are less effective when transferred to a related problem or during domain shifts (Zhao et al., 2020). One common method to overcome this limitation is to train a policy across multiple different tasks (Vithayathil Varghese & Mahmoud, 2020). In this setup, the agent explores each of the different tasks during training and then learns a single policy for all of these tasks simultaneously. One of the proposed benefits of this approach is that similar tasks could be learned faster as part of a single policy compared to learning a separate policy for each task individually. This is achieved by leveraging the experience from the first task to bootstrap the experience in the second task. While training a policy in multiple environments simultaneously can lead to a policy that can perform multiple tasks in one domain, it is yet to be understood how these learned skills can be transferred to a new set of environments or domains, as is common in other areas of machine learning. This paper aims to understand and present the design decisions and training protocols necessary to achieve such multi-task transfer learning of skills.

It has been shown that pre-trained models lead to large advances in transfer learning for Deep Learning, Computer Vision, and Natural Language Processing. Typically, the pre-trained models learn generalized representations through supervised (He et al. (2015), Krizhevsky et al. (2017)) or self-supervised learning (Devlin et al. (2019), Chen et al. (2021b), Chen et al. (2020), Grill et al. (2020)) on large amounts of data. Fine-tuning these pre-trained models then leads to efficient transfer for new downstream tasks (Chen et al. (2020), Grill et al. (2020), Devlin et al. (2019)). However, this technique cannot be directly applied to RL due to the large gap that may exist across different environments. While many simulation engines are flexible and can easily be used to emulate a diverse set of tasks and environments, the diversity of these tasks and environments leads to different state

and action spaces, and reward manifolds. This makes pre-training a general representation difficult. Thus, in the context of multi-task learning, policies for multiple tasks are typically learned *over a specific* environment, creating policies that cannot be easily transferred to a *completely new* environment with a similar set of tasks.

To overcome this gap in the efficient transfer of RL skills to new tasks and environments, this paper aims to study the design decisions and training protocols required for skill transfer and quantitatively evaluates their performance in Robotics environments in different domains. For an RL algorithm to be truly generalizable to similar tasks, it must also be able to apply different skills that it has learned in other environments with similar tasks. For example, learning to pick up a pan in a kitchen environment must be generalizable to picking up a toy in a bedroom. Recently, there has been advances in this direction with multitask transformers (Reed et al. (2022), Driess et al. (2023)), pre-trained image encoders for robotics (Nair et al. (2022), Parisi et al. (2022)), transformers trained on human demonstrations (Bousmalis et al. (2023), Brohan et al. (2023b)), or even data from internet sources (Brohan et al., 2023a). Each of these works provides an incremental step towards RL agents being able to apply their skills in a wide range of problem settings. While these works all do some form of multitask RL and transfer learning, they do not specifically explore the design decisions that enable the transfer of skills across different environments in a traditional RL setting.

An additional challenge impeding the general transfer of skills is the difference in state, action, and reward space that prevents any direct transfer as the policy can have different input and output sizes for each specific environment. This work aims to understand and enable the transfer of skills between environments with different robotic morphologies, where the state, action, and reward space differ. This environment-agnostic transfer is enabled by learning the appropriate latent representations through state and action translation layers, which map the states to a latent state space and then extract an action from latent space to an environment-specific action space, respectively. We demonstrate the effectiveness of these layers in continuous control tasks that transfer high-level manipulation concepts between the environments. Our experiments show that learning a policy in a shared, environment-agnostic latent (SEAL) space by translating states to this SEAL space and decoding latent skills into actions from the SEAL space yields sample efficiency gains anywhere from 7% to 72% compared to training on a single environment's tasks alone.

The proposed method, and thus the efficient skill transfer across domains can be achieved through minimal modifications in existing RL algorithms. We demonstrate skill transfer between the Meta-World (Yu et al., 2021) and Franka Kitchen (Gupta et al., 2019) environments, and show how SEAL can be applied to state of the art algorithms, such as Soft-Actor Critic (SAC) (Haarnoja et al., 2018).

Our contributions are as follows:

- We propose the shared, environment-agnostic, latent policy architecture, which allows the transfer of multiple skills across environments with different state and action spaces by using a single policy.
- We show that learning with our proposed architecture leads to learning specific tasks in up to 72% less policy network updates.
- We provide insights and limitations on traditional pre-training methods and how different design decisions may lead to limited exploration or skill transfer capabilities of the RL algorithm.
- We provide an open-source implementation to apply SEAL on existing RL algorithms.

## 2 PROBLEM STATEMENT

Reinforcement learning is formulated using a Markov decision process (MDP) (Sutton & Barto, 2018), where an MDP $\mathcal{M}$ is a tuple of $(\mathcal{S}, \mathcal{A}, P, r, \gamma, p)$, where $\mathcal{S}$ is the state space, $\mathcal{A}$ is the action space, the probability transition function $P : \mathcal{S} \times \mathcal{A} \rightarrow [0, 1]^s$ , $r : \mathcal{S} \times \mathcal{A} \rightarrow \mathbb{R}$ the reward function, $\gamma$ in $[0, 1]$ is the discount factor, and $p$ is the initial state distribution. At each time step, the agent can observe the state at time $t$, $\mathbf{s}_t$, choose an action sampled according to some policy $\pi : \mathcal{S} \rightarrow \mathcal{A}$ based on $\mathbf{s}_t$, receive a reward for landing in-state $\mathbf{s}_{t+1}$, and observe state $\mathbf{s}_{t+1}$. The goal of the agent is to find a policy that maximizes expected rewards for the current task $\mathbb{E}[R(\tau)]$ where $R(\tau)$ is the sum of rewards along the trajectory induced by following the policy $\pi$.

In the multitask reinforcement learning problem, a task distribution must be chosen where $N$ tasks are sampled from a task set $\mathcal{T}$ according to $t \sim p(t)$. Each task can be viewed as having its own MDP $\mathscr{M}_t$. Then, for task $t_i$ for $i$ in $1, 2, \ldots, N$, the MDP is $(\mathcal{S}, \mathcal{A}, P_i, r_i, \gamma, p_i)$ where the state space, action space, and discount factors are held constant across tasks while the probability transition function, reward function, and initial state distribution are specific to each task $t_i$. The goal of the multitask reinforcement learning agent is to maximize the expected reward across each task.

$$\mathbb{E}_{t \sim p(t)}[\mathbb{E}_{\tau \sim \pi}[R(\tau)]] \tag{1}$$

using policy $\pi_\eta(a|s,t)$, where $t \in \mathcal{T}$ is the task identifier and $\eta$ are learnable parameters of the policy.

In this paper, the problem we consider is extending multitask RL to multiple environments that don't share the same state and action spaces but are still related tasks. Let $\mathcal{E}$ be a set of $M$ environments, where each environment $m \in \mathcal{E}$ has a set of $\mathcal{T}_m$ distinct tasks. Each task $t_m \in \mathcal{T}_m$ in each environment $m$ has an MDP $\mathscr{M}_t^{(m)}$ for each task $t$. Therefore, the MDP $\mathscr{M}_j^{(i)}$ is now comprised of the tuple $(\mathcal{S}^{(i)}, \mathcal{A}^{(i)}, P_j^{(i)}, r_j^{(i)}, \gamma, p_j^{(i)})$, where the elements in the tuple can change with respect to the environment.

The environment and the task can be jointly sampled from a joint distribution $m, t \sim p(m, t)$. Since the state space $\mathcal{S}^{(m)}$ and action space $\mathcal{A}^{(m)}$ are environment-dependent, traditional multitask reinforcement learning algorithms cannot be applied to share policy parameters $\pi_\eta$ between tasks because the input state and output actions can be a variable number of sizes. In order to overcome this problem, we propose using translation layers to the input and output of the policy.

With the use of translation layers and action heads, our goals are two-fold. The first goal is to determine if we can demonstrate skills transfer across environments. The second goal is to find the best training approach to learn a single policy $\pi_\eta$ that can act in all environments. We propose this solution in Section 3.

## 3 ENABLING SKILL TRANSFER

In this section, we present the methods that can be used for multi-task skill learning and transfer. Further, in Section 4, we evaluate the transfer of skills between environments as well as how a single policy can be learned across multiple environments using our SEAL policy. To this end, we mainly explore two different methods where the first method is a pre-train and fine-tune method, while the second method trains across both environments using a SEAL policy architecture inspired by the multi-head architecture of (Yu et al., 2021). We report the mean success rates for these experiments, with Appendix F containing network update data for all experiments.

### 3.1 BASELINE MULTI-HEAD REINFORCEMENT LEARNING

In the most basic setting for learning multiple tasks in multiple environments, a policy for each environment can be trained as shown in Figure 1a. In this case, we independently train a single policy for each environment $m \in \mathcal{E}$ using the multitask multihead SAC (MTMHSAC) algorithm from (Yu et al., 2021). It should be noted that there are no shared parameters or features between either policy, therefore an environment $i$ will have policy $\pi_\eta^{(i)}$.

### 3.2 PRE-TRAINING AND FINE-TUNING

For this method, we train on one environment using MTMHSAC as a pre-training phase (Section 3.1). We then fine-tune the pre-trained policy on the unseen environment using MTMHSAC. This scenario is depicted in Figure 1b. In order to pre-train and fine-tune the policies in this manner, we modify the state spaces to ensure that the input dimensionality of each network is the same by padding the inputs with zeros. We also align the locations of the goals, and one-hot task IDs across state spaces. In order to have the same dimensionality we then pad any needed elements of the state with zeros. The final layer of the policy is different between environments because of the difference

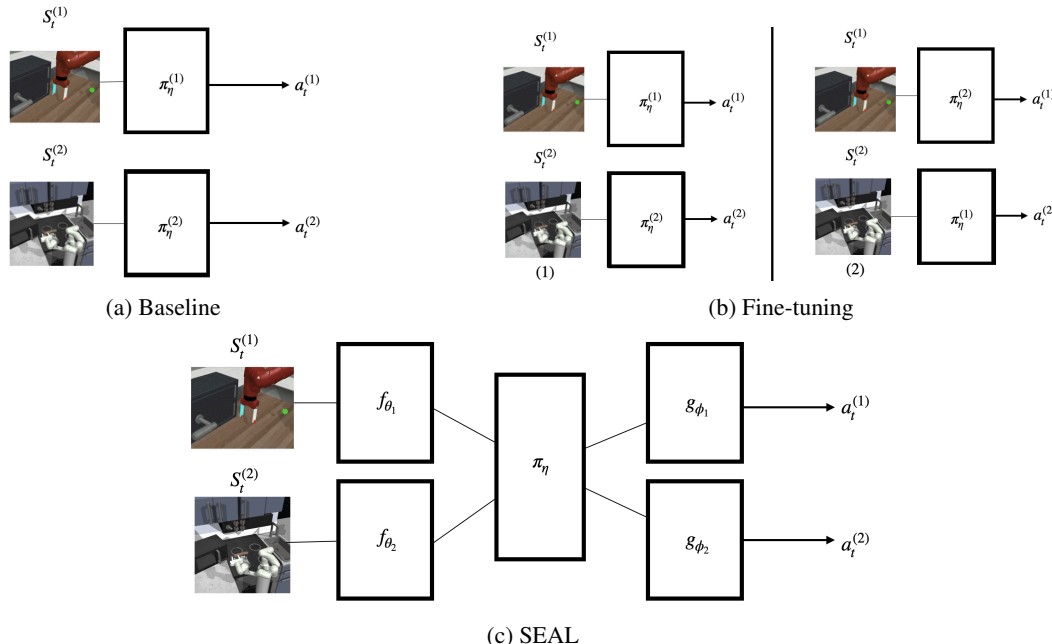

(a) Baseline

(b) Fine-tuning

(c) SEAL

Figure 1: Control architecture for multi-task learning. Top-Left: Typical multitask reinforcement learning architecture for single environments. We use this setup for generating our baseline success rates and sample complexities. Top-Right: Fine-tuning method. This method is a two-step process. In step (1) we pre-train policy $\pi_\eta^{(1)}$ on Meta-World and policy $\pi_\eta^{(2)}$ on Franka Kitchen. In step (2) we take the policy $\pi_\eta^{(2)}$ and fine-tune it on Meta-World, and fine-tune policy $\pi_\eta^{(1)}$ on Franka Kitchen. Bottom: The shared, environment agnostic, latent (SEAL) policy.

in action spaces. Because of this, we randomly initialize the final layer when fine-tuning. For pre-training, we train a policy $\pi_\eta^{(i)}$ on environment $i$ using MTMHSAC. We then take the policy and fine-tune it on environment $j, j \neq i$ with MTMHSAC.

### 3.3  SHARED, ENVIRONMENT AGNOSTIC, LATENT (SEAL) POLICY

To have an environment-agnostic policy that can learn in new environments while leveraging the skills previously learned, we propose the SEAL policy found in Figure 1c. Typically, different environments do not share the same action and state spaces. This issue is overcome by our proposed SEAL policy.

We resolve this issue of different state spaces by using a *state translation block* $f_{\theta_i}(\mathbf{s}_t^{(i)})$, where $\theta_i$ represents the parameters of the translation layer for the $i$-th environment and $\mathbf{s}_t^{(i)}$ the state at time $t$ for the $i$-th environment. Note that we assume that the task identifier is part of the state $\mathbf{s}_t^{(i)}$. We maintain a set of policy parameters denoted $\pi_\eta$ that are shared between all environments.

Finally, we must overcome the difference in action spaces. Similarly to the state translation block, we use different *action heads* $g_{\phi_i}$ for each environment indexed by $i$. The output size of an action head $i$ depends on the size of the action space of the environment $i$. Finally, for some environment $i$ to produce an action $a_t^{(i)}$ at time step $t$, a forward pass must be completed as $g_{\phi_i}\left(\pi_\eta\left(f_{\theta_i}(s_t^{(i)})\right)\right)$.

These parameters are then optimized using the objective $E_{t \sim p(t)}[E_{\tau \sim \pi}[R(\tau)]]$. However, because there are $M$ different environments, there are $M$ different optimizers, where an optimizer $i$ is operating on the translation layer $f_{\theta_i}$, the action head $g_{\phi_i}$, and the shared policy parameters $\pi_\eta$. Doing so allows the policy to share the high-level concepts present in each environment, such as pushing, grasping, and reaching, across all $M$ environments.

# 4 EXPERIMENTS

In this section, we present experimental results from the design decisions and training protocols required to achieve general skill representations that will then transfer to different tasks and domains. Following the evaluation procedure (Appendix B), we will establish a performance baseline and outline every design decision's impact and importance by comparing their performance gain against the established baseline. In the following, we will describe the training protocol steps and outline strategies for efficient skill transfer.

## 4.1 BASELINE MULTI-TASK EXPERIMENTS

The transfer capabilities are benchmarked using Meta-World (MW) and Franka Kitchen (FK) environments (see evaluation procedure in Section B). For the MW baseline, MTMHSAC is used due to its capabilities and the extendability of its implementation to our proposed network architecture (Yu et al., 2021). The baseline results are depicted in Figure 2, showing a success rate of about 75% using MTMHSAC. Like the MW baseline, MTMHSAC is used to establish the FK baseline. As the FK environments were not designed for a multi-task reinforcement learning setting but instead for hierarchical RL with sparse rewards, modifications to the reward functions were necessary to provide dense rewards for MTMHSAC (see Section C).

Figure 2 shows the baseline results of MW and FK. It is observed that the MW agent, on average, achieves a maximum success rate of 73%, and the FK agent reaches an average success rate of 74%. As can be seen from the baseline results in Figure 2, the Meta-World MTMHSAC agent reaches it's maximum performance in approximately a quarter of training, while the Franka Kitchen MTMHSAC agent reaches it's maximum performance in approximately half of training.

We believe that this is attributed to the fact that each task in MW is isolated to its own environment. Therefore, when training an agent, the agent does not need to explore and isolate the task to be solved from any distractions, such as the other tasks. This is not the same in FK, where all tasks are always available, which introduces an exploratory component of agent learning to FK. This observation has motivated the need for efficient skill transfer and the design of an algorithm that can leverage existing knowledge, and automatically infer and isolate tasks.

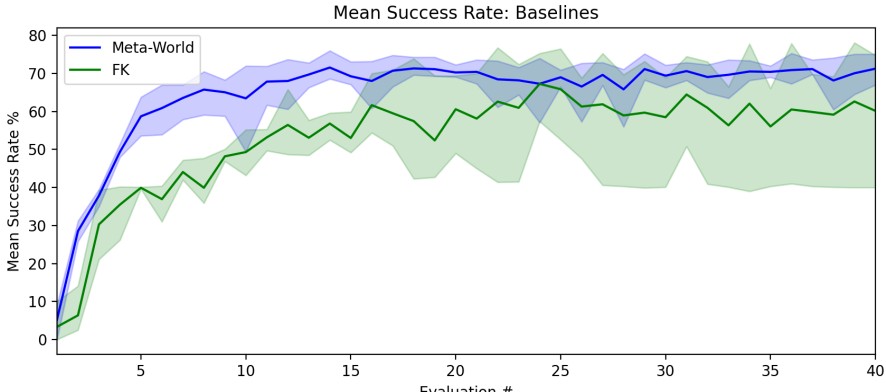

Figure 2: SAC results on Meta-World and Franka Kitchen. The shaded regions indicate the standard deviation.

## 4.2 FINE-TUNING PRE-TRAINED MODEL

Similar to traditional Machine Learning approaches, we first analyze the transfer capabilities of RL agents using the most straightforward approach: fine-tuning, where the policy is trained in one problem setting and then fine-tuned in a different problem setting. This training procedure is described in Appendix D.

We find that fine-tuning a pre-trained policy and value function limits the ability of an agent to use only the skills that were acquired in the pre-training stage, and additional skills are required through exploration, yielding no additional performance gains. In Figure 3, we show the performance of an agent pre-trained on FK and fine-tuned on MW. While we see no performance increase in terms of success rate, we do find that there are decreases in the number of samples required to learn specific tasks.

In Table 6, we report the number of network updates to reach 90% success rate in this experiment. This table shows that it took 29% less network updates for the Window Close task and 17% fewer network updates for the Window Open task to be learned. By examining the success plots in Figure 3, we observe that within a few fine-tuning steps, the agent is on par with training on MW alone based on the success rate. However, we can see a slight stagnation after a few evaluations. This stagnation is because of the limited number of skills that are available within FK. At this point, the agent must learn either new skills or how to leverage some combination of skills already learned while training on FK. The agent is able to leverage the skills it acquired during pre-training on FK, as shown by the decrease in network updates, to learn the Window Open and Window Close tasks.

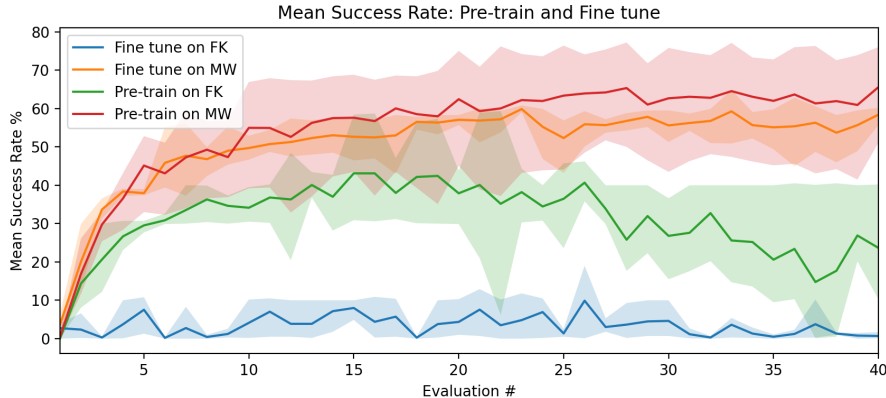

Figure 3: Pre-training and fine tuning success rates for each scenario. The shaded regions indicate the standard deviation.

When pre-training on MW, the easier of the environments, then fine-tuning on FK, we find that the fine-tuning process is limited in success rate because the MW value functions have overfit. In Figure 3, we report the success rate of the agent during the pre-training phase and the fine-tuning phase. We also report in Figure 7 the learning curve for the value functions in two experiments; fine tuning on MW and fine tuning on FK. Figure 7 shows that the value functions are likely overfit during fine tuning on FK. The pre-training phase shows that the MW agent reaches about 70% success rate. During the fine-tuning phase, the maximum success rate of the agent is $\sim 10\%$, while the number of network updates that it takes to learn Slide Cabinet is 25% less than learning in FK alone. We believe that when a pre-trained policy is fine-tuned in MW, the overfit value functions limit the ability for the agent to transfer it's learned skills.

One of the limitations of this pre-train and fine-tune approach is that we are left with two policies, one from the pre-training phase and one from the fine-tuning phase. The goal of multitask RL is to learn a single policy that can accomplish many different tasks. In Section 4.3 we explore how to learn a single policy across multiple environments.

## 4.3 SHARING ENVIRONMENT-AGNOSTIC LATENT REPRESENTATIONS

The main challenges of learning a single policy across multiple environments are the differences between the state space $\mathcal{S}$, and the action space $\mathcal{A}$, as well as dealing with various reward landscapes. To overcome these challenges, we use the network architecture that was outlined in Section 3.

In order to evaluate our SEAL policy architecture, we first train on the MW and FK environments simultaneously. We find that a limited number of tasks can be solved in this experiment, as shown in Figure 4, where the success rate approaches 40%. We found that the Slide Cabinet task is learned

in $\sim 75\%$ fewer network updates, showing that there is a limited transfer of skills in this experiment. While it is not inherently obvious, the drawer close and the slide cabinet tasks are fairly similar. Both tasks require the agent to learn to move the robotic arm towards the object, place the robotic arm on the object in a manner that allows a force to be applied to the object, and apply a force to the object to complete the task. Based on the number of updates to learn the slide cabinet task, it seems that a transfer of skills is occurring, but it could also be that the policy is learning each skill individually.

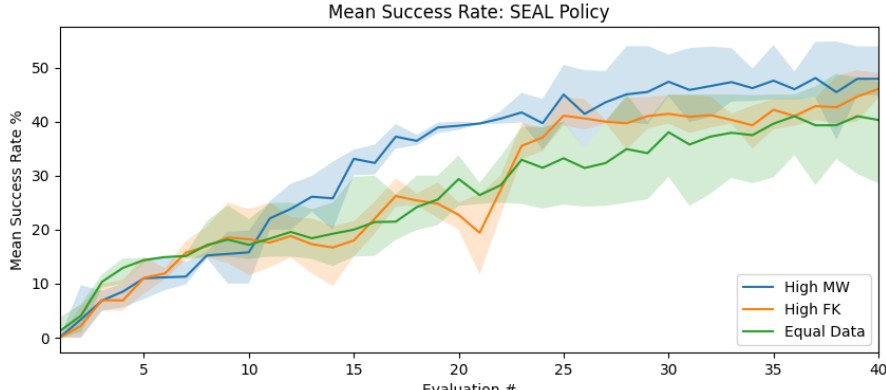

Figure 4: Success rate when training on MW and FK simultaneously. There are three variants of training on both MW and FK: (a) equal data ratio (green), (b) high MW data ratio (blue), and (c) high FK data ratio (orange). The shaded regions indicate the standard deviation.

We have designed the following experiment in order to examine whether the SEAL policy transfers skills or learns the tasks individually. For the first 1000 epochs of training, the agent receives $\sim 90\%$ of data from environment $i$ and $\sim 10\%$ from environment $j$. For the next 1000 epochs of training, the agent receives $\sim 80\%$ of data from environment $i$ and $\sim 20\%$ from environment $j$. This process of manipulating the data ratios continues in the same manner until the ratio reaches $50\%$ data for both environments.

By limiting the amount of data from an environment that the agent uses to optimize its policy and value function, we find that the agent is able to transfer the learned skills. First, we examine the experiments where the agent receives a high amount of FK data and a low amount of MW data at the start of training. In Table 1, we report the number of network updates, in millions, needed to reach 90% success rate on certain tasks during training. In the High FK column, we report that when training with a high ratio of data from FK we find the following improvements in comparison to training on FK or MW alone: drawer close needs 60% less network updates, drawer open needs 50% less network updates, while slide cabinet needs 55% less updates, and microwave needs 32% less updates.

Next, we find similar skills transfer in the experiments where the agent receives a large amount of MW data and a low amount of FK data at the start of training. The number of network updates to learn specific tasks is reported in the High MW column of Table 1. When training with a high ratio of MW data, we find that the slide cabinet needs 55% less network updates, and the microwave needs 32% less network updates compared to learning on MW or FK alone.

Both of these results are indications that a transfer of skills is occurring. We believe that this improvement in the number of network updates to learn a skill is because in the previous experiment each environment had an equal opportunity to modify the shared latent policy embedding space. The improvement in sample complexity in regards to each task shows us that our architecture is effectively capturing the skills that can transfer, while also aligning the outputs of the translation layers for related tasks. With these two modules, the action head for each environment is then able to learn how to produce actions that apply the desired skill based on the latent skill embedding that the shared latent policy embedding produces.

Table 1: Millions of network updates for each training procedure. Bolded results indicate a decrease in number of network updates for that task to reach 90% success rate.

| Task | Single Env | FK&MW (50/50 Data) | High FK | High MW |
|------|-----------|-------------------|---------|---------|
| Drawer Close (MW) | 6.4 | 10.7 | **2.4 (-62.76%)** | 11.5 |
| Drawer Open (MW) | 42.7 | 211.2 | **24.8 (-42.0%)** | 53.9 |
| Reach (MW) | 17.1 | 19.2 | **12.7 (-25.88%)** | 27.9 |
| Slide Cabinet (FK) | 14.9 | 19.2 | **11.5 (-23.0%)** | **4.3 (-71.0%)** |
| Microwave (FK) | 76.8 | 147.2 | **38.3 (-50.13%)** | **71.5 (-6.94%)** |

## 5 RELATED WORK

One of the first comprehensive benchmarks for testing in the multitask RL domain was Meta-World (Yu et al., 2021). The benefit of using the Meta-World benchmark is that it has a high degree of shared environment and control structure which allows for efficient learning of distinct but related tasks (Yu et al., 2021). Another benefit of Meta-World is the dense reward function available for each individual task. (Yu et al., 2021) propose several benchmark algorithms for multitask reinforcement learning including Multi-Task Multi-Head Soft Actor Critic (MTMHSAC). The MTMHSAC algorithm modifies the base Soft-Actor Critic algorithm by adding an entropy head for each task, allowing for different levels of exploration per task (Yu et al., 2021). In Yu et al. (2021) the states are augmented with a one-hot vector that indicates which environment the state belongs to. Other recent approaches include Soft-Modularization which uses the one-hot vector as input to a routing network that outputs probabilities for how data is routed through the policy network (Yang et al., 2020), Yu et al. (2020) developed a method of projecting conflicting gradients onto the same plane thus making network optimization more efficient, Cho et al. (2022) developed a variational based method as well as a measure of negative transfer. Recently, He et al. (2023) have shown diffusion models to be effective planners and data synthesizer in multitask reinforcement learning settings.

Gupta et al. (2019) originally designed Franka Kitchen as a benchmark for algorithms that can solve long-horizon, multitask problems. Franka Kitchen was then modified by (Fu et al., 2020). The robotic arm in Franka Kitchen is a 9-DOF Franka robot that is placed in a kitchen environment with many different household kitchen objects. The goal of the environment is to achieve some desired configuration of the objects through manipulation (Gupta et al., 2019).

There has been a recent interest in creating foundation models Bommasani et al. (2022) for different domains in machine learning. These foundation models provide a model that can be easily fine-tuned to some downstream task (Bommasani et al., 2022). These types of pre-trained models have led to rapid advances in deep learning for computer vision and natural language processing. These types of models can be train in either a supervised learning (He et al., 2015) or unsupervised (Devlin et al., 2019) fashion. The rapid advances can be attributed to the generalized representations that these pre-trained models have learned to extract or generate; they are typically trained on large diverse datasets (Krizhevsky et al. (2017), Lin et al. (2015), Pennington et al. (2014), Mikolov et al. (2013)).

These advances in pre-trained models have not been as useful to the RL community as they have to the wider machine learning community. Recent work has found that pre-trained computer vision models can be used as a frozen encoder module for downstream reinforcement learning (Parisi et al., 2022), while advances in natural language processing have led to the use of transformers (Vaswani et al., 2023) in reinforcement learning (Chen et al. (2021a), Reed et al. (2022), Janner et al. (2021)) that show the capabilities of the transformer as a policy for continuous control RL. Driess et al. (2023) showcased the abilities of a vision language model for continuous control tasks where both images and text are inputs to the transformer that can then output high-level tasks for the reinforcement learning agent.

Finally, Reed et al. (2022) was one of the first transformer-based model to have the vision-language transformer model output actions for continuous control tasks directly. One limitation of Reed et al. (2022) is that the effectiveness of the trained model in real-world robotic manipulation tasks is limited. Meanwhile, Lee et al. (2022) aimed to enable effective generalization capabilities by training on suboptimal data from many Atari games. Brohan et al. (2023b) show the effectiveness and scaling capabilities of a transformer-based reinforcement learning agent on real-world tasks through many human demonstrations, while Brohan et al. (2023a) shows the effectiveness of using multiple data

sources, including internet data, for real-world robotic manipulation tasks across different robotic arm configurations. Thus, we have seen a step taken in the direction of foundation models for RL, but these works do not answer the question of how to transfer skills across multiple environments, especially in an online setting.

One line of work related to ours is the multi-embodiment of RL agents. These works attempt to learn different policies that can be adapted to any number of different robot morphologies, different environment dynamics, or some combination of both (Gupta et al., 2022). A number of different approaches have emerged in the literature including using graph neural networks (Kurin et al. (2021), Wang et al. (2018)), shared modular policies per embodied joint (Huang et al., 2020), and transformers (Gupta et al., 2022). Our work is different from this line work because we attempt to (a) learn multiple skills across the different robot morphologies, (b) learn across different proprioceptive state sizes, and (c) our set of skills to learn is more diverse.

The challenge to learn diverse skills across different morphologies by identifying task-specific and robot-specific modules in the learned parameters has been conducted in (Devin et al. (2016), Gupta et al. (2017)). Devin et al. (2016) proposed a method of composing different robot skills by creating robot morphology modules, as well as task modules. These combinations of task and robot morphology modules are then trained simultaneously and can then be recombined for new, unseen tasks (Devin et al., 2016). One limitation of this work that our method attempts to overcome is the simultaneous training of each robot morphology on a task. In other domains of machine learning, it is possible to train a model on one dataset and fine tune that model to a new dataset in a few gradient steps. Gupta et al. (2017) proposes a method that uses contrastive loss to learn proxy tasks across environments via a shared feature space. This shared feature space can then be used to transfer additional knowledge from one robot to another. In addition to these works, Hausman et al. (2018) learn an embedding space for skills where a skill vector contains information about the skill to be applied which is then concatenated to observations. Hausman et al. (2018) found that they were able to interpolate between skills for zero shot task completion. D'Eramo et al. (2020) trained a similar model to ours, except they chose environments that only contain a single skill to transfer to other environments. Our work attempts to expand on the limited number of skills that were tested in previous works. While there are several different types of robot included in Gupta et al. (2022), there is a limited number of skills transferred across the different morphologies, and each of the skills were to be transferred. Our work contains a larger number of potential skills to transfer.

## 6 CONCLUSION

This work focused on the ability of multi-task Reinforcement Learning agents to transfer their learned skills to new environments. In order to enable skill transfer across environments, we first had to overcome the issues of different state and action spaces for reinforcement learning environments. Our first method of overcoming this issue was to pad and align the state spaces of different environments such that an agent could be trained in one environment and then use the trained agent parameters as a starting point for learning in a new environment. This approach showed that the fine-tuning of a pre-trained agent can have some success in a new environment, however there were limitations based on the skills and exploration learned in the pre-training environment. Next, we attempt to learn a single policy across diverse environments. We propose the shared, environment-agnostic, latent (SEAL) policy for this task. This SEAL policy has environment-dependent parameters along with shared parameters. The shared parameters were used in order to capture any shared skills between the two environments, while allowing for the environment-dependent parameters to handle the differences in state and action spaces. Over two diverse continuous control environments, Meta-World and Franka Kitchen, we show that the SEAL policy combined with an effective training protocol can decrease the number of network updates to learn a task by up to 72%. One of the limitations of our work is that we only utilize state spaces. An extension of our work would be to extend this method to reinforcement learning from pixels, or to explore the ability to transfer skills across robots of different morphologies or using orthogonal skills. Another interesting line of research may be to investigate why the Meta-World agent's are overfit, and how to regularize those agents in a way that maintains performance.

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

## A BACKGROUND INFORMATION

### A.1 REINFORCEMENT LEARNING

In this work, the Reinforcement Learning (RL) algorithm that we use is Soft-Actor Critic (SAC). SAC is an off-policy RL algorithm that is based on the maximum entropy learning framework (Haarnoja et al., 2018). This framework leads the agent to maximize both expected rewards and entropy, which leads the agent to succeed in the task while acting as randomly as possible (Haarnoja et al., 2018). In this work there are three sets of parameters to optimize: the soft Q-function $Q_\theta(s_t, a_t)$ with $Q$ parameterized by $\theta$, the policy parameters $\pi_\phi(a_t|s_t)$ where $\pi$ is parameterized by $\phi$, and the entropy penalty coefficient $\alpha$ (Haarnoja et al., 2018). The objective for policy optimization is:

$$J_\pi(\phi) = \mathbb{E}_{s_t \sim D}[\alpha \log \pi_\phi(a_t|s_t) - Q_\theta(s_t, a_t)] \tag{2}$$

with $\alpha$ controlling the entropy penalty coefficient. The coefficient $\alpha$ is learned using the objective:

$$J(\alpha) = \mathbb{E}_{a_t \sim \pi_\phi}[-\alpha \log \pi_\phi(a_t|s_t) - \alpha\bar{\mathcal{H}}] \tag{3}$$

where $\bar{\mathcal{H}}$ is the minimum target entropy.

Previous work has modified the SAC algorithm to include an entropy term for each of the $N$ tasks to guide exploration in each task individually, as well as to introduce replay buffers per task (Yu et al. (2021), Yang et al. (2020)).

### A.2 META-WORLD

The first set of environments used in this work are from Meta-World (Yu et al., 2021). Meta-World is a suite of multitask RL and meta-RL environments that consist of a number of robotic manipulation tasks. These environments are subdivided into different sets with any number of environments and different goals. This work focuses on the Multi-Task 10 (MT10) set of environments. In the MT10 set, there are 10 tasks that the RL agent can interact with. Some environments are closely related to each other, such as window open and window close, while other environments are not as closely related to each other, such as pick and place, and window close. This difference in tasks allows for a wide variety of skills to be learned across these environments with robust learning happening due to the number of goals available for each of the individual tasks(Yu et al., 2021). Meta-World also provides a dense and smooth reward function to help RL agents learn(Yu et al., 2021). The different tasks of the MT10 set of environments can be found in Figure 5.

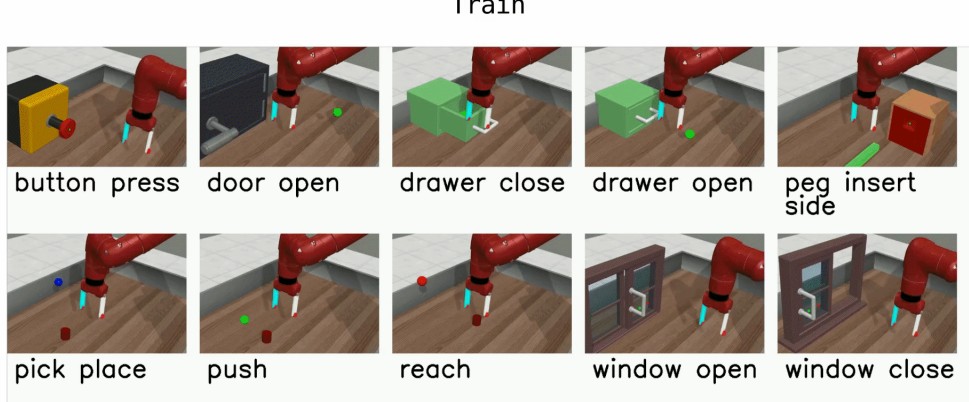

Figure 5: Meta-World MT10 tasks, image from (Yu et al., 2021).

## A.3 Franka Kitchen

The Franka Kitchen environment is the other environment to be used in this work. The Franka Kitchen environment has typically been used in hierarchical RL where the goal is to complete a number of tasks sequentially (Gupta et al., 2019). This work uses the Franka Kitchen environment in a slightly different manner where each of the available tasks in Franka Kitchen are used individually to create a similar set of tasks to MT10 from Meta-World. This can be verified in Figure 6. Thus, the agent only needs to solve one of the available tasks in a single environment. The default reward function in Franka Kitchen is a sparse reward function where the agent only receives a reward for solving the task (Gupta et al., 2019).

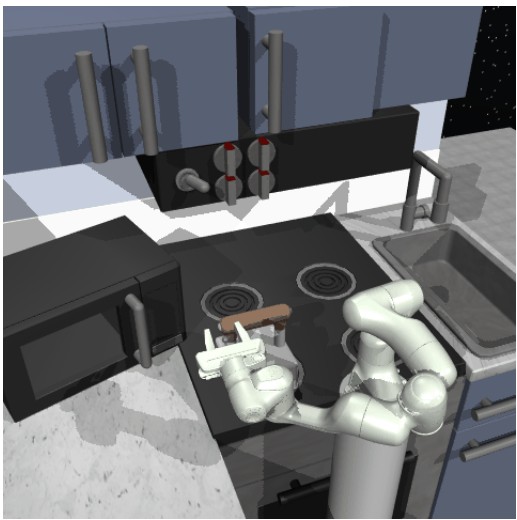

Figure 6: Franka Kitchen environment, image from (Gupta et al., 2017).

## B    Evaluation procedure

We evaluate the performance and transfer capabilities of the RL agent in two domains: Meta World (Yu et al., 2021) and Franka Kitchen (Gupta et al., 2019)(see Section A.2 and A.3). In all experiments, we report the success rate across 100 episodes per task with 50,000 gradient steps each to ensure that our results are statistically significant. These results are reported across 2 different random seeds. For our multitask policy $\pi_\theta(a|s,t)$, the task identifier $t$ is a one-hot encoding of the task to inform the agent what task it is solving.

In addition to the success rate, we also report the number of gradient update steps to our policy that it takes to learn specific tasks. In order to calculate the number of updates it takes to learn a task, we choose a success threshold of $90\%$, once a policy learns a task past this threshold for the first time we can then calculate the number of network gradient updates it took to learn this task. This value is calculated by the following formula: $C * GS * BS$, where $C$ is the current epoch, $GS$ is the number of gradient steps per epoch, and $BS$ is the number of samples of data for this task.

## C    Reward function for Franka Kitchen

The Franka Kitchen environment uses a sparse reward function that gives the agent a reward of 0.3 for completing the desired task and 0 otherwise. This limits the ability to do online RL as it is extremely difficult for the agent to learn the correct sequence of actions to complete any of the tasks. To overcome this challenge in creating our baseline approach, we adopt a modified version of the dense reward function that is used in Meta-World for Franka Kitchen. Table 2 shows the mapping between the Franka Kitchen task to be solved and the reward function used from Meta-World to provide dense rewards.

Table 2: Dense reward functions used for Franka Kitchen from Meta-World.

| Franka Kitchen task | Meta-World Reward Function |
|---|---|
| Microwave | Door Open |
| Right Hinge Door | Door Open |
| Left Hinge Door | Door Open |
| Light Switch | Sweep |
| Top Right Burner | Dial Turn |
| Top Left Burner | Dial Turn |
| Bottom Right Burner | Dial Turn |
| Bottom Left Burner | Dial Turn |
| Slide Cabinet | Push |
| Kettle | Pick Place |

Some slight modifications are made to the reward functions to use them in Franka Kitchen. Both Meta-World and Franka Kitchen were designed using Mujoco. However, the objects that a RL agent interacts with in Meta-World are Mujoco bodies, while most of the Franka Kitchen objects are attached to bodies. The exception to this in Franka Kitchen is the kettle as it is a body itself. To overcome this issue, the reward functions for each task are modified slightly to use different Mujoco sites of the object of interest. For example, the microwave reward function uses the handle site as a method of determining how open or closed the door is. We refer the interested reader to our public implementation for further details on how the reward function was modified to use sites. To denote that a task has been completed, we used a threshold of 5 cm.

## D    FINE TUNING EXPERIMENTS DETAILS

To ensure that we could fine tune pre-trained policies and value functions across environments, there are some extra steps that had to be taken. The first step is to ensure that state and action spaces had the same dimensionality. In order to do this, we have padded the Meta-World states with zeros to match the dimensions of the Franka Kitchen state. We also had to pad the Meta-World actions with zeros to match the dimensions of the Franka Kitchen actions. Initially, we naively pad the states with zeros for the last dimensions, leaving the state intact. We find that this led to limited performance because when we transfer to a new environment the order of the input state vector was different from what the policy and value functions are trained on. In order to align these input features we spliced the state, one-hot ID, and goal together in the same method across the environments. We also find that aligning the one-hot IDs across environments was important to our success. In our original experimentation, we did not assign similar tasks with equal one-hots IDs. We show the alignment of tasks across environments with the same one-hot IDs in Table 3. Once we did this alignment, along with the padding of state spaces, we were able to fine-tune the pre-trained policy and value-functions effectively as outlined in our results.

Table 3: Environments that were aligned across MW and FK are shown. The Button Press Topdown, Window Close, Peg Insert Side, Reach, Drawer Open, Top Right Burner, Top Left Burner, Bottom Right Burner, and Bottom Left Burner tasks all received unique one-hot IDs.

| Franka Kitchen task(s) | Meta-World task(s) |
|---|---|
| Slide Cabinet | Drawer Close |
| Kettle | Pick Place |
| Left Hinge Door, Right Hinge Door, Microwave | Door Open |
| Light Switch | Push |

To pre-train an agent in an environment, we first modify the state space as mentioned above. Once this step is completed, we use MTMHSAC to pre-train the agent for 10000 epochs. During this training, the agent is evaluated using the evaluation procedure outlined in Appendix B. Once the

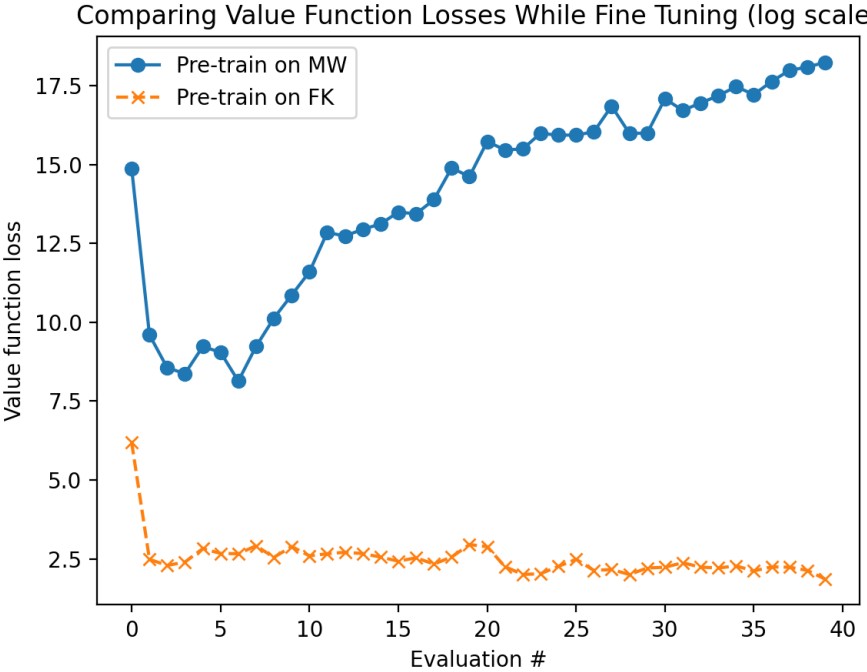

Figure 7: The mean loss of the value functions when fine tuning on FK or MW.

agent has been trained, we then load the agent's parameters that had the highest success rate during pre-training as a starting point in the new environment. All parameters are trained, while the last layer of the policy is randomly initialized because this layer is directly related to the action size of an environment.

## E  IMPLEMENTATION AND COMPUTE DETAILS

Code will be open-sourced upon acceptance.

## F  NETWORK UPDATES DATA

Table 4: Millions of network updates for each training procedure on Meta-World tasks. This number is based on how many network updates it took to get to a 90% success rate. 0's indicate that the task was not learned in that training configuration.

| Task | Single Environment | FK&MW(50/50) | High FK | High MW |
|---|---|---|---|---|
| Door Open | 32.0 | 168.5 | 46.9 | 69.1 |
| Drawer Close | 6.4 | 10.7 | 2.4 | 11.5 |
| Drawer Open | 42.7 | 211.2 | 24.8 | 53.9 |
| Window Open | 25.6 | 185.6 | 46.9 | 72.7 |
| Window Close | 14.9 | 155.7 | 43.7 | 64.8 |
| Peg Insert Side | 0.0 | 0.0 | 0.0 | 0.0 |
| Pick Place | 0.0 | 0.0 | 0.0 | 0.0 |
| Push | 0.0 | 0.0 | 0.0 | 140.8 |
| Button Press Topdown | 49.1 | 215.5 | 91.7 | 110.9 |
| Reach | 17.1 | 19.2 | 12.7 | 27.9 |
| Total Network Updates | 187.7 | 966.4 | 269.1 | 551.7 |

Table 5: Millions of network updates for each training procedure on Franka Kitchen tasks. This number is based on how many network updates it took to get to a 90% success rate. 0's indicate that the task was not learned in that training configuration.

| Task | Single Environment | FK&MW(50/50) | High FK | High MW |
|------|--------------------|--------------|---------|---------|
| Slide Cabinet | 14.9 | 19.2 | 11.5 | 4.3 |
| Microwave | 76.8 | 147.2 | 38.3 | 71.5 |
| Top Right Hinge Cabinet | 0.0 | 0.0 | 0.0 | 0.0 |
| Top Left Hinge Cabinet | 125.9 | 243.2 | 0.0 | 0.0 |
| Top Right Burner | 19.2 | 0.0 | 0.0 | 0.0 |
| Top Left Burner | 185.6 | 0.0 | 0.0 | 0.0 |
| Bottom Right Burner | 6.4 | 0.0 | 0.0 | 68.4 |
| Bottom Left Burner | 138.7 | 0.0 | 0.0 | 0.0 |
| Kettle | 0.0 | 0.0 | 0.0 | 0.0 |
| Light Switch | 14.9 | 125.9 | 110.1 | 41.2 |
| Total Network Updates | 582.4 | 535.5 | 159.9 | 185.4 |

Table 6: Millions of gradient updates for pre-training on Meta-World, or pre-training on Franka Kitchen then fine-tuning on Meta-World. The pre-train on MW policy is then fine-tuned on Franka Kitchen for Table 7. This number is based on how many network updates it took to get to a 90% success rate. 0's indicate that the task was not learned in that training configuration.

| Task | Single Environment | Pre-train on MW | Fine-Tune on MW |
|------|--------------------|-----------------|-----------------|
| Door Open | 32.0 | 25.6 | 59.7 |
| Drawer Close | 6.4 | 6.4 | 6.4 |
| Drawer Open | 42.7 | 106.7 | 209.1 |
| Window Open | 25.6 | 44.8 | 21.3 |
| Window Close | 14.9 | 23.5 | 10.7 |
| Peg Insert Side | 0.0 | 0.0 | 0.0 |
| Pick Place | 0.0 | 0.0 | 0.0 |
| Push | 0.0 | 238.9 | 0.0 |
| Button Press Topdown | 49.1 | 117.3 | 91.7 |
| Reach | 17.1 | 102.4 | 102.4 |
| Total Network Updates | 187.7 | 665.6 | 501.3 |

Table 7: Millions of gradient updates for pre-training on Franka Kitchen, and pre-training on Meta-World and then fine-tuning on Franka Kitchen. The pre-train on FK policy is then fine-tuned on Meta-World for Table 6. This number is based on how many network updates it took to get to a 90% success rate. 0's indicate that the task was not learned in that training configuration.

| Task | Single Environment | Pre-train on FK | Fine-tune FK |
|------|--------------------|-----------------|--------------|
| Slide Cabinet | 14.9 | 12.8 | 134.4 |
| Microwave | 76.8 | 185.6 | 0.0 |
| Top Right Hinge Cabinet | 0.0 | 0.0 | 0.0 |
| Top Left Hinge Cabinet | 125.9 | 200.5 | 0.0 |
| Top Right Burner | 19.2 | 38.4 | 179.2 |
| Top Left Burner | 185.6 | 0.0 | 0.0 |
| Bottom Right Burner | 6.4 | 72.5 | 0.0 |
| Bottom Left Burner | 138.7 | 0.0 | 0.0 |
| Kettle | 0.0 | 0.0 | 0.0 |
| Light Switch | 14.9 | 14.9 | 21.3 |
| Total Network Updates | 582.4 | 0.0 | 334.9 |

