# OpenReview forum: "Understanding the Transfer of High-Level Reinforcement Learning Skills Across Diverse Environments"
_ICLR.cc/2024/Conference — Submitted to ICLR 2024_

### Official Review · Reviewer_8Y8K · 2023-10-29

**Soundness:** 2 fair
**Presentation:** 3 good
**Contribution:** 3 good
**Rating:** 5
**Confidence:** 3

**Summary:**

The authors tackle the problem of transferring skills across environments with different  states and action spaces using learned embeddings. To this end, the authors first motivate need for their design decision of a shared policy architecture by analyzing traditional pre-traiing methods and individual training baselines. Based on these insights, the authors propose a shared policy architecture with state and action embeddings to tackle heterogenous spaces across environments. They demonstrate improved sample complexity when transferring skills between Meta-World (MW) and Franka’s Kitchen (FK) environments.

**Strengths:**

## Originality
The problem of transferring skills across heterogeneous spaces has been previously tackled by works in this subfield. The proposed architecture is novel to the best of my knowledge.

## Quality
The work is generally decent, albeit with some issues mentioned in the weakness section.

## Clarity
The paper is generally written straightforwardly with not a lot of typos. The ideas are easy to understand, and the impact of the core contributions is not hard to grasp.

## Significance
The work seems to be approaching a complete state, and to me, the finding regarding the connection between data mixture and skill transfer seems to be the most exciting. Slightly more analysis, such as visualizing the learning dynamics of SEAL (Policy oscillation mentioned by the authors) and the impact of skill diversity, could significantly boost the significance of this work.

**Weaknesses:**

While I think the paper is written in a sound manner, I believe it still needs improvements in the Experiments, analysis, and formatting:

## Experiments
- Experiments across 2 random seeds are not enough in my opinion. Generally, we try to evaluate a statistically significant number of runs and report either the mean + deviation or the IQM and deviation [Aggarwal et. al, 21]
- The authors propose multiple explanations to why the data mixture in the SEAL network has an impact. Their explanations, unfortunately, seem to be to be more on the intuitive level and not expanded upon enough. While it seems evident that changing the data mixture improves skill transfer, the claim regarding oscillations in policy parameters should be quite easy to justify and demonstrate in the experimental setup of authors.
- The authors do not mention their architectural and training hyperparameters. Additionally, the design decisions for evaluations are not substantiated (100 episodes, 50,000 gradient steps) i.e. I do not understand how the authors decided for the given evaluation protocol. (see Questions section). My recommendation would be to substantiate this by either specifically mentioning the sources of previous architectures that the authors used as a reference, or providing a rationale for future reproducibility.

## Formatting
- In table 1, I do not understand the N/A columns. I had to separately look for them in the appendix in Tables 4--6. I would recommend mentioning these in the text in the fourth paragraph of section 4.3
### Section 2:
- Do the authors mean that the policy outputs a probability distribution over actions from which one can sample actions? Or do they mean that action space lies in [0,1]?
- Trajectories are never introduced mathematically. I would recommend adding a small definition
What is p(t) -- the probability of t? This is confusing
- If t is used as task identifier, then do the authors mean a new task is sampled at each timestep? If so, please add a small sentence clarifying this. Or use a seperate simple for task identification
### Section 3.3:
- Do the state embeddings and action heads share the same parameters ? If not, use a separate symbol.
- I would recommend formalizing this as an algorithm to make the training procedure for SEAL clearer.
- Apart from the shared architecture, is the data mixture from the two environments the only design decision? If not, I would recommend summarizing all of them in this section for better clarity

**Questions:**

- In section 4.1, the authors mention that the reason for the MTMHSAC agent reaching the maximum reward in FK is the presence of other tasks as distractors. Do they believe this causes negative gradient interference? If so, shouldn’t gradient surgery during fine-tuning fix this? [Yu et al., 2020] Additionally, do the authors consider approaches such as gradient surgery relevant to their architecture?
- Are there connections between this work and the work of Hausman et al. and Deramo et al.?
- How does the diversity of skills in the source environment impact the performance? There seems to be some form of overfitting prevalent in the value functions. However, given that the data mixture is a significant design decision, I am wondering if this could be dependent on the choice of the environment as the source and transfer.
- What causes the overfitting of the value functions in MW? How can one mitigate this?
- Do the authors believe different neural architectures enable some form of scaling laws for more complex environments?
- In Appendix D, the authors mention the necessity to align the task IDs between MW and FK. Is the alignment of task IDs related to incorporating relational structure between similar tasks? [Mohan et al., 2023]  If so, Has this been a common practice in previous works?


1. [Yu et. al, 2020] Yu, T., Kumar, S., Gupta, A., Levine, S., Hausman, K., & Finn, C. (2020). Gradient surgery for multi-task learning. Advances in Neural Information Processing Systems, 33, 5824-5836.
2. [Aggarwal et. al, 21] Agarwal, R., Schwarzer, M., Castro, P. S., Courville, A. C., & Bellemare, M. (2021). Deep reinforcement learning at the edge of the statistical precipice. Advances in neural information processing systems, 34, 29304-29320.
3. [Hausman et. al.] Hausman, K., Springenberg, J. T., Wang, Z., Heess, N., & Riedmiller, M. (2018, February). Learning an embedding space for transferable robot skills. In International Conference on Learning Representations.
4. [Deramo et. al] D'Eramo, C., Tateo, D., Bonarini, A., Restelli, M., & Peters, J. (2019, September). Sharing knowledge in multi-task deep reinforcement learning. In International Conference on Learning Representations.
5. [Mohan et. al., 2023] Mohan, A., Zhang, A., & Lindauer, M. (2023). Structure in Reinforcement Learning: A Survey and Open Problems. arXiv preprint arXiv:2306.16021.

---

> ### Author Response · Authors · 2023-11-16
> **Reply to reviewer 8Y8K (1)**
>
> Dear reviewer 8Y8K. Thank you for taking the time to review our paper. We do agree with your comments about the paper in regards to the formatting, and the number of random seeds. For submission we were only able to perform experiments over 2 random seeds, however we have updated the manuscript plots and tables to reflect experiments over 3 random seeds.
>
> In regards to our evaluation protocol we made those changes for two reasons which we have add to the paper. Thank you for mentioning this as it makes the evaluation procedure clearer. The first reason is to ensure statistical significance of our results. When evaluating our agent, we are sampling from a distribution so to eliminate variations in our evaluations we increased the number of evaluations to 100, and limited the evaluation frequency to every 50,000 gradient steps. It also had the extra benefit of speeding up our result collection, as our codebase uses a multi-processing sampler that caused significant overhead when evaluating the agent more frequently. We will be releasing our codebase upon acceptance of our work, as reproducibility of our results is one of our top priorities as well.
>
> The authors propose multiple explanations to why the data mixture in the SEAL network has an impact. Their explanations, unfortunately, seem to be to be more on the intuitive level and not expanded upon enough. While it seems evident that changing the data mixture improves skill transfer, the claim regarding oscillations in policy parameters should be quite easy to justify and demonstrate in the experimental setup of authors.
>
> Yes, the oscillations explanation is very intuitive and not backed by any experiments. We apologize as that should not have been in the manuscript. We have removed that point from the updated manuscript.
>
> In section 4.1, the authors mention that the reason for the MTMHSAC agent reaching the maximum reward in FK is the presence of other tasks as distractors. Do they believe this causes negative gradient interference? If so, shouldn’t gradient surgery during fine-tuning fix this? [Yu et al., 2020] Additionally, do the authors consider approaches such as gradient surgery relevant to their architecture?
>
> Maybe we could have been more clear. The multi-task reinforcement learning agent does not reach its maximum success rate in Franka Kitchen as quickly as Meta-World does. This is because in Franka Kitchen all tasks are available at all times. Thus the agent must isolate the task in the environment before solving the task, which is what we meant as distractors. There could be negative gradient interference, but that is not what we intended to say nor have we conducted any experiments that indicate that.
>
> In regards to the comment about gradient surgery, we do not think at this point that gradient surgery would be related. What would most likely be more relevant is some kind of “reward surgery.” What we mean by this is the following: for each task we have a reward function that is dense and smooth. We also have any number of tasks that have some overlap in the skills required to solve them. An example of this would be the button press topdown task in Meta-World and the light switch task in Franka Kitchen. In order to align these tasks more effectively for transfer, one idea may be to design the reward functions of these two tasks in a more similar manner, until the point that the tasks diverge. However there are no measures that we know of that can decompose two tasks in this manner so reward functions can be aligned. If that decomposition is possible, we might then build reward functions that share meaningful similarities that enable more efficient policy learning between tasks.

---

> ### Author Response · Authors · 2023-11-16
> **Reply to reviewer 8Y8K (2)**
>
> Are there connections between this work and the work of Hausman et al. and Deramo et al.?
>
> We did miss these works in our literature review, thank you for pointing them out.
> We have updated the related work section accordingly. We outline the differences between our work, Hausman et al, and Deramo et al below.
>
> The work of Hausman et al describe an embedding module trained to capture a skill ID that is then concatenated to the policy’s observation, whereas we attempt to capture our embedded skills in our policy. Hausman et al. attempt to learn skills on one robot morphology and transfer those skills on the same robot to new tasks via interpolation between skills, in a pseudo-meta-learning approach. Where our work differs from Hausman et al is that we are learning in an online fashion across both robotic morphologies simultaneously. We do not attempt to interpolate between skills but instead capture the skill in the SEAL policy parameters. We have evidence of these skills being captured when examining the number of network updates required to learn in this simultaneously learning setting.
>
> The work of Deramo et al could be seen as an initial instantiation of our work. Our work expands on the number of skills learned across environments while using a similar architecture. Deramo et al do learn from pixels, whereas we learn from states. The work of Deramo et al chooses environments which have similar tasks, much like our work, but there is only one task per environment to transfer from. In our work we attempt to transfer multiple skills from one environment to another. Our architectures are very similar, although we have chosen continuous action space environments whereas Deramo et al used environments with discrete action spaces. Therefore Deramo et al were able to use a Deep Q Network algorithm, while we used a Soft Actor Critic algorithm.
>
> How does the diversity of skills in the source environment impact the performance? There seems to be some form of overfitting prevalent in the value functions. However, given that the data mixture is a significant design decision, I am wondering if this could be dependent on the choice of the environment as the source and transfer.
>
> The overfitting of the value function mentioned in the manuscript is when pre-training the RL agent on Meta-World. Performance when fine-tuning on Franka Kitchen is due to this overfitting, which would be related to a lack of plasticity in the value function (and/or policy) at that time [https://arxiv.org/pdf/2305.15555.pdf]. We have updated the paper to make this overfitting point more clear.
>
> The diversity of skills could limit the performance in terms of overall success rate. When comparing the performance plots across the three simultaneous training regimes we experimented with in comparison to learning on Meta-World or Franka Kitchen alone, it is clear that there is a drop in success rate. However, our goal was not to generate SOTA results on either environment, instead our goal was to find effective training methods and an architecture that enables the transfer of skills across differing state and action spaces. Therefore we did not chase higher performance. Given enough time and compute, it may be possible to produce results with superior performance across tasks.
>
> What causes the overfitting of the value functions in MW? How can one mitigate this?
>
> From our experimentation alone it is not clear what is the cause of the overfitting in Meta-World. It could be due to the lack of exploration in Meta-World where each task is isolated in its own environment, while in Franka Kitchen all tasks are available in the environment. It could be related to this fact as well as the Meta-World agent learns more quickly than the Franka Kitchen agent, then the Meta-World policy and value functions are repeatedly fit to the sampled data without any improvement which would lead to overfitting. This question may lead to an interesting question for future work: how can we regularize policy and q-function learning in multi-task reinforcement to avoid overfitting while achieving reasonable performance? We’ve added this idea to the paper as a potential for future work.
>
> Do the authors believe different neural architectures enable some form of scaling laws for more complex environments?
>
> We haven’t examined this in particular but this could be one of the design decisions that enables our transfer of skills. We have noticed in recent work (RT-1, RT-2, RT-X-1, PaLM-e), that part of the reason for their improvements in success is the scaling up of data and model size. For the experiments where we attempt to transfer skills, we did increase the data and model sizes. However it is unclear whether this trend would continue for N > 2 environments. It could be something that we examine in more detail in a follow up work where we attempt to learn on N > 2 environments on pixel based reinforcement learning problems.

---

> ### Author Response · Authors · 2023-11-16
> **Reply to reviewer 8Y8K (3)**
>
> In Appendix D, the authors mention the necessity to align the task IDs between MW and FK. Is the alignment of task IDs related to incorporating relational structure between similar tasks? [Mohan et al., 2023] If so, Has this been a common practice in previous works?
>
> Yes! In our initial experimentation, not included in the manuscript, we found that pre-training & fine-tuning policies and value functions with unaligned state spaces led to very limited performance. This is because we trained the policy & value function on one distribution of states with a one semantic meaning of where the goal, one-hot ID, and state information is. Then when transferring to the other environment the policy and value functions had to learn a new distribution of state space with different semantic meanings for each of the elements of the state. To overcome this we attempted to incorporate some relational structure in the one-hot task IDs, where we aligned similar tasks (ie slide cabinet and drawer close had the same one-hot task ID) and dissimilar tasks had their own one-hot task ID. We also aligned the states to have the same dimensionality via padding the smaller state space to the size of the larger state space with zeros, and to have goal positions in the same position in the state space across each environment. We have not seen this approach taken in prior work, as to the best of our knowledge we are the first to tackle this problem in multi-task reinforcement learning with differing state and action spaces. Thus our work is the first to incorporate one-hot task IDs.

---

> ### Comment · Reviewer_8Y8K · 2023-11-21
>
> Thank you for the reply!
>
> - **Formatting:** Thank you for considering these changes. I would like to see them in the revision.
> - **Evaluation Protocol:** Thank you for additionally reporting the third seed. I appreciate the authors taking reproducibility seriously. I want to restate why I mentioned the issue regarding seeds and statistical significance. -- As mentioned in Aggarwal et al. and many previous works cited in the paper, there is a common tendency for RL algorithms to overfit the seeds that they are being trained and evaluated upon. Therefore, how the experiments are reported can seriously impact the perception of the result. Hence, I recommended attending IQMs across at least 5 seeds (In Figure 4 of the Meta-world paper, the evaluation was done on 10 seeds, although Aggarwal et. al argues for higher), specifically mid-50 % IQMs. I would appreciate it if the authors included these in the revision.
> - **Oscillation and Data Mixture:** The point regarding the impact of data mixtures is interesting. My original critique was of how the explanation was delivered and the lack of empirical grounding. While I see that the authors might need help to run further experiments to validate the post hoc explanations in this rebuttal phase, I want to mention that this discussion does add to the novelty. I suggest refining the write-up in the revision or further drafts.
> - **Task distractors:** Thank you for clarifying the point regarding gradient surgery. I also suggest explaining in the revision how distractors affect the reward function in both environments. The explanation provided in the response is a pretty good candidate.
> - **Related Work:** Thank you for including these. In agreement with reviewer Um91, I recommend that the authors scan the Multi-Task and Meta-RL literature again to ensure they cover the related work sufficiently. The papers I mentioned were just two examples of this line of work. Surveys by [Beck et al., 2023](https://arxiv.org/abs/2301.08028) and Mohan et al. could be good references. I can only comment on the comprehensiveness of this part once the related work has been updated in the revision.
> - **Overfitting:** Thank you for clarifying the point regarding overfitting. I look forward to seeing the updated explanation in the revision.
> - **Relational Inductive bias:** I recommend adding the data that motivated this design decision in the appendix. As the authors mentioned, the relational structure helps mitigate the need to learn a new distribution of state space with different semantic meanings for each state element in the new task. Since this is not standard practice, I think it becomes essential to comment on whether this is a central design decision in the performance benefit or a way to mitigate sample inefficiency. Given further computation, do the authors expect the SEAL agent to be able to learn these distributions as well? Does not having this relational bias fundamentally limit the learning capability?
> - **Latent representation:** I agree with reviewer awgj that the quality of the learned latent representation should be evaluated. I would recommend including this analysis in the revision.

---

### Official Review · Reviewer_Um91 · 2023-10-30

**Soundness:** 1 poor
**Presentation:** 2 fair
**Contribution:** 1 poor
**Rating:** 3
**Confidence:** 4

**Summary:**

The paper proposed a way to do multi-task reinforcement learning on different environments. The method is described as pretraining on one environment and then finetuning on a different environment with a environment-specific state encoder layer and action decoder layer, as well as a shared policy in the latent state-action space. The method is tested on Meta-world and the Franka Kitchen environments and the results show that the proposed method enables the agent to learn with fewer gradient update steps.

**Strengths:**

1. Transferring between different environments instead of different tasks for one environment is an interesting and important problem setting.

2. The proposed method is generally easy to follow.

**Weaknesses:**

1. For transferring between different environments, the authors propose to first pretrain on one environment and finetune on another environment, which I believe is a common approach that has been used in many recently proposed multitask RL methods. Another contribution the authors list is that the proposed method allows the transfer between tasks with different state and action space. The proposed method is to learn a task-specific state encoder and action decoder to deal with input size, which I believe is also a common way already used in practice. Therefore, I think the novelty of this paper is limited.

2. The paper emphasizes the transfer of "skills" while I don't think the standard formulation for skills/options are mentioned in the method. The paper is more related to multi-task reinforcement learning and meta-reinforcement learning, for which I think the authors should include more discussions in the related work section.

3. More experiments need to be done regarding the efficiency and asymptotic performance of the proposed method. The authors only show the comparison of the number of network updates comparison on only four different tasks in MetaWorld and Franka Kitchen.

4. I don't think the number of network updates is a good metric for evaluating the sample efficiency of a multi-task RL method.

**Questions:**

See Weaknesses.

---

> ### Author Response · Authors · 2023-11-16
> **Reply to reviewer Um91**
>
> Dear reviewer Um91. Thank you for taking the time to review our work and for bringing up some weaknesses of our work. We would like to address your comments below.
>
> “For transferring between different environments”:
> We have conducted a literature search and to the best of our knowledge we have not found any other multitask reinforcement learning approaches that pre-trains and fine-tunes, nor have we found any work that learns a task specific state encoder and action decoder for learning in an online fashion across environments with different state and action spaces. Could you please provide us with the relevant papers/citations so we can add them to our work?
>
> The paper emphasizes the transfer of "skills":
> We do agree that there are different ways to formulate the problem, such as with options. The focus of our work is to learn skills in order to solve tasks in a certain environment while having the learnt policy be able to apply those learned skills in another environment with similar tasks. Our results show that this is possible thanks to our SEAL architecture, with no changes to the underlying reinforcement learning algorithms.
>
> We would like to address Weakness 3 & 4 together.
> More experiments need to be done regarding the efficiency and asymptotic performance of the proposed method. The authors only show the comparison of the number of network updates comparison on only four different tasks in MetaWorld and Franka Kitchen.I don't think the number of network updates is a good metric for evaluating the sample efficiency of a multi-task RL method.
> Our goal was not to generate SOTA results on Meta-World or Franka Kitchen. The goal was to investigate the situations in which transfer can occur.
>
> This number of updates metric shows how the SEAL architecture can transfer skills because the SEAL policy contains latent skills from one of the environments that can easily be decoded and applied to a new environment. Success rate alone would not be an interesting metric because there would be no difference if the policy solved a task on the first evaluation or the last. With this metric we have a quantifiable way to show that a transfer of skills has occurred, once the task can be solved at a 90% success rate. What additional experiments would you have liked to see? What metric would you recommend to use in place of our network updates metric?

---

> > ### Comment · Reviewer_Um91 · 2023-12-03
> >
> > **"To the best of our knowledge we have not found any other multitask reinforcement learning approaches that pre-trains and fine-tunes"**. I believe this is not true. In fact, a lot of recent papers follow the paradigm of pretraining on multiple tasks and finetuning on a new task. Examples: Meta-q-learning (Fakoor et al.), Deep online learning via meta-learning:Continual adaptation for model-based rl (Nagabandi et al.). The authors may also want to check Gato (Reed et al.) for transferring between tasks with different states and action spaces.
> >
> > **"The focus of our work is to learn skills in order to solve tasks in a certain environment while having the learnt policy be able to apply those learned skills in another environment with similar tasks"**. The authors keep mentioning "skills" in the title, abstract and main text but I still can't find a detailed formulation for it. Is it just the policy network? Is it temporally extended? Or is it just the mapping to the latent representation?
> >
> > **"What additional experiments would you have liked to see? What metric would you recommend to use in place of our network updates metric?"**. 1. The authors choose MetaWorld (Yu et al.) as the testing environment, which has over 50 different manipulation tasks. However, the authors only pick three of them and evaluate their algorithm. Thus it is hard to identify how general the conclusions are. 2. I understand the goal is to investigate the situations when transfer can occur. But if one method can eventually achieve a success rate that is much higher than the other but with a few more network updates, would you still suggest that the other method is better? Besides, after reaching one success rate, in practice the performance of RL methods might oscillate a lot or even drop. Is that also described by your metric? In general, I would suggest the authors to consider both the number of network updates and the asymptotic performance when evaluating transferring.

---

### Official Review · Reviewer_awjg · 2023-10-30

**Soundness:** 2 fair
**Presentation:** 2 fair
**Contribution:** 2 fair
**Rating:** 3
**Confidence:** 5

**Summary:**

This paper studies transfer and multi-task RL with the goal of transferring a learnt policy from one task to another. The key contribution is the study of how to design policies from existing well known RL algorithms, such as soft actor critic (SAC) that can be transferred across tasks. The paper studies transfer learning across tasks by learning a shared embedding space, that allows transfer to be achieved even if the state or action spaces across tasks differ.

**Strengths:**

1. To achieve transfer across tasks, the work proposes to use minimal modifications to existing RL algorithms, and studies this on popular benchmark domains. Additionally transfer learning can be achieved even if the state or action space changes across tasks, due to the ability to learn a latent policy architecture.

2. The proposed minimal approach to study transfer learning can be applied directly on top of existing algorithms, with the code to be released for wide adaptation.

3. The paper is quite simple to understand with the key contribution of the paper written clearly. I like that there are no overclaims made by the authors, and the approach is very minimalistic, if not completely novel.

4. The paper studies a broad and challenging problem of transfer : which is how to achieve transfer across tasks when the state and action spaces change; a lot of works have shown transfer when the reward functions change, or if mid-way in the learning process, the reward changes, but previous works do not explicitly account for if transfer can be achieved when the state spaces differ.

5. Transfer is achieved through the latent representation, where the states are first mapped to a latent, and then actions are decoded from the latent space depending on the action space of the task. The authors term the latent space as the skills space, or the SEAL space, where the latents are represented through the policy network, which can be environment agnostic.

**Weaknesses:**

I think the biggest limitation or challenge of the approach is through the architectural approach of achieving transfer learning itself. A lot of prior works have studied this sort of shared latent space architecture, or learning of latents, such that the latents can be transferred across tasks. This is not completely new; and the paper instantiates this in one particular way, keeping simplicity in mind through existing RL frameworks.

My biggest worry with such works is the ability to learn a good latent representation, one that can recover the structure of the task, and makes it useful to be shared across tasks. In other words, how can the authors qualitatively and quantitively understand whether this shared SEAL space is good or bad for transfer? Ideally, the latent should capture the underlying dynamics of the environment.

It would have been useful if more details could be provided on the approach section; Section 3 can perhaps be expanded better; I understand the need for simplicity for the proposed approach, but in its current form, it seems there is not really much algorithmic novelty in the work. The multi-heaed SAC approach is nothing new either; and the authors dont really provide much details on the learnt latent representations. For example, do we need a separate/different representation objective? Is this reward free representation objective?

Experimental results are too naive, with very few results and not really providing an exhaustive understanding of the proposed approach. Results section can clearly be improved (and dont really need this big figure plots perhaps?)

**Questions:**

1. Can the authors show some qualitative results showing how good this learnt latent representation is? How do we know these latents are good for transfer across tasks?

2.  do not mean to see performance plots, such as cumulative returns - rather I am looking for more results showing how well the structure of one environment can be learnt; how good latents can be captured from the task, and what would make them useful for transfer?

3. I am inclined to believe that this sort of approach may perhaps work well, assuming good latent representation can be learnt, in pixel based tasks, instead of raw state/action spaces. This is because often for pixel based environments, the underlying structure of the environment can be recovered - can the authors demonstrate some results, using for example a simplistic CNN based SAC with a shared embedding space, that can be used for transfer?

4. I like the problem statement and the need for a simplistic approach; but I think the authors can do a much better job at this and I would encourage the authors to do; for example, if you can study different representation objectives, task specific reward based or even reward free, and then show which of the proposed objectives can learn a good latent representation that enables transfer across tasks - this would be really interesting! I think the algorithmic novelty of the work is not really there; so rather the authors can turn this into an empirical validation paper studying the ability of what makes good representations to be transferred across tasks. This can be done for simple to complex control environments, ranging from raw state/action to even pixel-based environments.

---

> ### Author Response · Authors · 2023-11-16
> **Reply to reviewer awjg**
>
> Dear reviewer awjg. Thank you for your time in reviewing our paper. We would like to address your highlighted weaknesses/questions below:
>
> Can the authors show some qualitative results showing how good this learnt latent representation is?
>
> We are currently investigating methods in which we can show this.
>
> I am inclined to believe that this sort of approach may perhaps work well, assuming good latent representation can be learnt, in pixel based tasks, instead of raw state/action spaces. This is because often for pixel based environments, the underlying structure of the environment can be recovered - can the authors demonstrate some results, using for example a simplistic CNN based SAC with a shared embedding space, that can be used for transfer?
>
> This is definitely a limitation of our current work that we should have mentioned in the submitted manuscript. Unfortunately given our time and compute budget it is unrealistic to add these experiments to the current version of this manuscript. However this was already in our plans for the next iteration of this work. We thank the reviewer for this comment and we have updated the manuscript to note this limitation in the Conclusion section.
>
> I like the problem statement and the need for a simplistic approach; but I think the authors can do a much better job at this and I would encourage the authors to do; for example, if you can study different representation objectives, task specific reward based or even reward free, and then show which of the proposed objectives can learn a good latent representation that enables transfer across tasks - this would be really interesting! I think the algorithmic novelty of the work is not really there; so rather the authors can turn this into an empirical validation paper studying the ability of what makes good representations to be transferred across tasks. This can be done for simple to complex control environments, ranging from raw state/action to even pixel-based environments.
>
> We thank the reviewer for this comment. We agree with the reviewer on this comment: that our approach is very simplistic. However, that was our goal. We show that with no changes to the underlying reinforcement learning algorithm, we can achieve a transfer of skills across environments with differing state and action spaces thanks to our SEAL policy architecture. Other approaches that are tangentially related to ours add extra objectives in order to do their transfer, or they condition the output action on the type of robot arm. In our work we don’t do any of this in order to provide a method that can work without any modifications or extra objectives. Future work could study the addition of these representation learning or reward free objectives, however that is not within the scope of this work which we framed as more of an exploratory work. We had planned to extend this work along the lines suggested by the reviewer as part of future work, and we will incorporate your suggestions as part of future work. We also incorporated the rest of your suggestions into our paper.

---

### Official Review · Reviewer_NkEv · 2023-11-01

**Soundness:** 2 fair
**Presentation:** 1 poor
**Contribution:** 1 poor
**Rating:** 3
**Confidence:** 5

**Summary:**

This paper explores several approaches to transferring high-level RL skills across diverse environments with different state and action spaces. The goal is to study the effect of skill transfer on multi-task sampling efficiency. Concretely, they compare three methods: (1) a simple baseline that learns a separate policy for each environment, (2) a pretraining-finetuning method that learns a policy in one domain and finetunes on the other, where any mismatch in state space is bridged by padding, and (3) training an environment-agnostic policy with a shared latent backbone (SEAL) on data across multiple environments. They found that (1) works reasonably well, (2) suffers from suboptimal performance when finetuned in downstream environments, and (3) achieves some level of skill transfer across environments. Moreover, the sample efficiency of (3) can be improved by adjusting the ratio of data from each environment. To summarize, this paper takes a step toward understanding the effect of transferring high-level skills across environments on multi-task RL.

**Strengths:**

- The problem setting studied in this paper is of importance to the community. Achieving data sharing across environments with different state and action spaces is a step towards generalist agents.
- The architecture of the shared environment-agnostic latent policy is rather intuitive.

**Weaknesses:**

- The second baseline method with pretraining + finetuning makes little sense. I'm not sure how padding the observation space can enable skill transfer, especially since the order of the state space is not restricted (i.e. dimensions corresponding to end-effector positions in one environment might represent object position in the other).
- The experimental results are largely inconclusive. There is no direct comparison between the three methods studied in the paper. Even if we combine the individual plots, it seems that skill transfer leads to suboptimal performance compared to training separately in each environment.
- The paper does not compare to external baselines from prior work.
- The presentation of the experiment section is rather disorganized.
- Overall, I don't think this paper demonstrates the level of rigor required for a conference paper.

**Questions:**

- How does padding the states enable data sharing when there is a mismatch in the semantic meaning of each state dimension?
- How does SEAL work when the environments are more different, say when one environment consists of a legged robot while the other involves a tabletop manipulator?

---

> ### Author Response · Authors · 2023-11-16
> **Reply to reviewer NkEv**
>
> Dear reviewer NkEv. Thank you for reviewing our work. We would like to address some of the weaknesses that you have pointed out, then answer your questions, and lastly summarize the changes we have made to the paper thanks to your feedback.
>
> “The second baseline method with pretraining + finetuning”: our goal with these experiments was to show that the pre-train & fine-tune paradigm seen in other areas of ML research is not very effective here, even if we take steps to align the state spaces in a semantically meaningful manner. We do attempt to align the values in the state spaces for these experiments. We have modified the paper to make this padding process more clear.
>
> “The experimental results are largely inconclusive”: our goal was not to generate SOTA success rates in Meta-World or Franka Kitchen. The goal of our work was to investigate and understand where and when skills may or may not transfer. We do report results that show our architecture does lead to transfer learning in this multi-task reinforcement learning problem. We prove this by calculating how many samples are needed to reach a 90% success rate with our architecture. We found that 5 tasks, reaching, drawer close, drawer open, slide cabinet, and microwave, need less samples when training on both Meta-World and Franka Kitchen using our SEAL policy architecture using the different training data ratios.
>
> “The paper does not compare to external baselines from prior work”: This is correct. To the best of our knowledge we are the first to produce work that investigates the issue of differing state and action spaces in a multi-task reinforcement learning problem setup. Therefore there were not any baselines to compare our work to. There are papers that are in this line of work however none of them deal with this issue of differing state and action spaces in a multi-task reinforcement learning problem. We’ve added these papers to the related work section of our paper. To apply those methods to our problem would have meant redesigning the previous methods as most related previous work deals with pixel based reinforcement learning where we deal with state based.
>
> We would now like to answer your questions:
> How does padding the states enable data sharing when there is a mismatch in the semantic meaning of each state dimension?
>
> At first, we did naively pad the observations with zeros preserving none of the semantic meaning of each position. This led to limited performance in those preliminary experiments that we did not include in the manuscript. To overcome this issue we padded the observations more intelligently - specifically we wanted to ensure that goals, and one-hot task IDs were in the same position in both observations. The included results for the pre-train and fine-tune experiments use this alignment of goals and one hot IDs.
> 	In addition to this, we also attempted to align the tasks across environments via the one hot IDs. For example, in our experiments we found that the drawer close and slide cabinet tasks were similar. Thus for the pre-train and fine-tune experiments we ensured that the observations from these tasks had the same one hot ID. The appendix of the paper outlines how the remaining tasks were aligned. We have modified the paper to include a description of this alignment process in the main text.
>
> “How does SEAL work when the environments are more different, say when one environment consists of a legged robot while the other involves a tabletop manipulator?”
>
> We did consider this problem setting as well. We picked this problem setting because we felt that it would be the most straightforward to investigate first. With what we have learned from this work we plan on investigating skill transfer of different robotic morphologies, and orthogonal skills, such as locomotion and manipulation.

---

> > ### Comment · Reviewer_NkEv · 2023-11-20
> >
> > Thanks for clarifying the details of the padding method. I think this baseline is much more convincing given the state and action dimensions are semantically matched. However, I disagree with the statement that there is no external baseline to compare to. While it may be true that no prior work solves this exact state action mismatch problem in multitask learning, it is possible to simply add environment-specific encoders to existing multitask learning methods and compare to your method. I also maintain my concern regarding the inconclusiveness of the empirical results. Even if we only look at the sampling efficiency (Table 1), it seems that joint training with equal data leads to worse sampling efficiency than training on each task individually. So getting the method to work may require careful tuning of the dataset balance. Moreover, there are more than 40 tasks in Metaworld and Franka Kitchen, but only 5 environments enjoy an improvement in sampling efficiency from joint training. With all these concerns unaddressed, I will maintain my current score.

---

> ### Author Response · Authors · 2023-11-20
>
> Dear Reviewer NkEv,
> Thank you for responding to our comments. We would like to further address your concerns.
>
> Inconclusiveness of Empirical Results
>
> That is correct, that training on both environments with the same amount of data limits the overall success rate and amount of transfer. That is why we introduced the experiments where we manipulate the data ratio. These experiments show the benefits of the SEAL architecture, where there is an improvement in how many network updates the policy requires in order to learn skills with minimal data.
>
> Number of Meta-World Environments
>
> We don't use all of the available environments in Meta-World, we only use the MT10 set. We chose the MT10 environments because after inspection of the available tasks we believed that they align well with the available tasks in Franka Kitchen. We will make this more clear in the manuscript.
>
> Lack of External Baselines
>
> We disagree with the reviewer that we could have made changes to existing works to overcome the issue of differing state and action spaces. To the best of our knowledge, we are the first to try and overcome this limitation. In addition, we are also the first to frame this problem in a multi-task reinforcement learning problem setting. Because of this, we are the only work that has an approach that is applicable in this multi-task reinforcement learning with differing state and action spaces problem setting. Therefore we produced our own baselines following the pre-training and fine-tuning paradigm that is prevalent in different areas of the machine learning. These baselines attempt to naively solve the problem, which our SEAL architecture then improves on.

---

### Meta-Review · Area_Chair_jhe8 · 2023-12-05

**Metareview:**

This paper evaluated different approaches for transfer across different environments, such as fine-tuning and learning a shared representation. Overall, it was agreed upon that the paper is not ready for publication. Different suggestions were made by the reviewers on how to improve the paper and I recommend the authors to use them at their discretion. Two crucial points for this decision were the fact that the main novelty of the paper is supposed to be in its empirical evaluation but that still falls short. In such a paper, understanding the learning dynamics or visualizing the learned representations to characterize their similarities would be quite important, instead of mostly relying on performance curves to draw conclusions. Finally, a key limitation of the paper is that the results are fairly inconclusive given that the results were over only 3 independent runs. It is hard to expect the reported results to generalize given how little data is being considered. That, by itself, is a deal breaker.

**Justification For Why Not Higher Score:**

As it is written in the meta-review, two crucial points for this decision were the fact that the main novelty of the paper is supposed to be in its empirical evaluation but that still falls short. In such a paper, understanding the learning dynamics or visualizing the learned representations to characterize their similarities would be quite important, instead of mostly relying on performance curves to draw conclusions. Finally, a key limitation of the paper is that the results are fairly inconclusive given that the results were over only 3 independent runs. It is hard to expect the reported results to generalize given how little data is being considered. That, by itself, is a deal breaker.

**Justification For Why Not Lower Score:**

N/A

---

### Decision · Program_Chairs · 2024-01-16

Reject